# Failure to replicate the diabetes alleviating effect of a maternal gluten-free diet in non-obese diabetic mice

Mia Øgaard Mønsted[ID], Laurits Juulskov Holm, Karsten Buschard, Martin Haupt-Jorgensen[ID]*

Department of Pathology, Rigshospitalet, The Bartholin Institute, Copenhagen, Denmark

* martin.haupt-joergensen@regionh.dk

## Abstract

Type 1 diabetes (T1D) is an autoimmune disease with an unexplained rising incidence for which environmental factors like gluten may play a role. Previously, we showed that a gluten-free (GF) diet provided strictly *in utero* reduces the autoimmune diabetes incidence in Non-Obese Diabetic (NOD) mice compared to a gluten-containing standard (STD) diet. The current study was initiated to elucidate possible mechanisms behind the diabetes-alleviating effect of the same diet intervention. NOD mice received either a GF Altromin diet or a STD Altromin diet during pregnancy. Female offspring from both groups were fed a STD diet throughout life and their diabetes incidence was recorded for 200 days. The following parameters were measured in 13-week-old female offspring: insulitis degree, glucose and insulin tolerance, and plasma insulin autoantibody titer. The diet intervention showed no reduction in autoimmune diabetes incidence, insulitis degree, glucose nor insulin tolerance and plasma insulin autoantibody titer. In conclusion, this study could not replicate the previously observed diabetes alleviative effects of a maternal gluten-free diet in NOD mouse offspring and could therefore not further elucidate potential mechanisms.

**Data Availability Statement:** All relevant data are within the paper and its Supporting information files.

## Introduction

Type 1 diabetes (T1D) is an autoimmune disease caused by destruction of the pancreatic beta cells resulting in hypoinsulinemia and hyperglycemia. There has been a steep increase in the T1D incidence during the past decades [1], which may be driven by environmental factors of unknown origin. Intake of gluten has been widely studied in relation to T1D. This is, among other reasons, due to a higher prevalence of celiac disease in T1D patients than in the general population [2] and the shared genetic risk haplotypes between these two diseases [3].

Several studies in both Non-Obese Diabetic (NOD) mice and BioBreeding rats have shown diabetes-alleviating effects of gluten-free (GF) diets, both when provided prenatally and during weaning, after weaning, and throughout life [4–7]. The time of introduction to gluten seems to affect the risk of T1D in humans, as seen from the Diabetes and Autoimmunity Study in the Young (DAISY), a prospective birth cohort study, which report that early introduction to gluten (<4 months of age) is associated with progression from islet

**Funding:** The author(s) received no specific funding for this work.

**Competing interests:** The authors have declared that no competing interests exist.

autoimmunity to manifest T1D [8]. Additionally, in the Type 1 Diabetes Prevention and Prediction (DIPP) prospective birth cohort study, gluten intake during early childhood (<6 years of age) was associated with increased risk of islet autoimmunity [9]. In a study investigating the timing of gluten introduction relative to T1D risk, we showed a marked decrease in autoimmune diabetes incidence from 62.5% to 8.3% in NOD mice on a gluten containing standard (STD) diet vs a GF diet provided strictly *in utero* (from conception to birth) [10]. In a later study, using the same experimental set-up and diet, our laboratory confirmed the reduced level of insulitis at prediabetes with a GF diet *in utero*, as well as an alleviative effect of the associated celiac disease [11]. This study also found reduced jejunal enteropathy, reduced local and systemic inflammation, and increased number of islets in the prediabetic NOD mice on a GF vs a STD diet *in utero*. This indicates that the mechanisms behind the diabetes-alleviating effect of a GF diet are complex and involves both the intestine and the islets. In fact, low-grade intestinal inflammation [12], increased intestinal permeability [13], and altered microbiota [14] are found in T1D patients, among other pathologies, making it a multifactorial disease.

The current study was initiated to further explore the mechanism behind the reported diabetes-alleviating effect of a GF diet *in utero*. Thus, we compared NOD mice on a GF diet *in utero* with NOD mice fed a lifelong gluten-containing STD diet.

## Methods

### Statement of ethics

The study was approved by the Danish Animal Experiments Inspectorate, Ministry of Food, Agriculture and Fisheries (license number 2016-15-0201-00841).

### Animal study design

6-week-old male and female NOD/ShiLtJ mice (The Jackson Laboratory, Bar Harbor, ME, USA) arrived at the specific-pathogen-free (SPF) animal facility at the Department of Experimental Medicine (University of Copenhagen, Copenhagen, Denmark). The mice were acclimatized for one week on Altromin 1434 control diet 5c (Altromin, Lage, Germany), our experimental STD diet. Breeding pairs were assembled by housing two female mice together with one male mouse on either the STD diet or an Altromin 1434 modified GF diet 5d (Altromin, Lage, Germany). Diet compositions are published in Antvorskov et al. [10]. Immediately after birth the mice on the GF diet were shifted to the STD diet to ensure that the GF offspring were subjected to a GF diet strictly *in utero*. Only females from 1st generation offspring were used. 2–8 mice were housed per ventilated cage (12–14 air exchanges/h, temperature 22˚C, humidity 55±10%, 12 h light cycle) and they were weaned at 3 weeks of age. The mice had free access to bottled tap water and diet throughout the study. Mice with blood glucose ≥12 mmol/L prior to tissue extraction and tolerance tests were excluded. This study followed the ARRIVE guidelines.

### Diet gliadin content

The gliadin content in the GF and STD diets used in the current study and in 2 batches of a GF diet used in previous studies [11] were measured using the RIDASCREEN® Gliadin detection kit (R-Biopharm AG, Darmstadt, Germany) on diluted grinded diet pellets, adhering to the manufactures protocol.

### Diabetes screening

From 10 weeks of age, the NOD mice were screened weekly for hyperglycemia by blood glucose measurements on tail blood using a hand held glucose monitor (Abbott Diabetes Care, Alameda, CA, USA). The diagnosis of autoimmune diabetes was based on blood glucose levels ≥12 mmol/L measured 2 times within a 2-day interval. Animals were immediately killed by cervical dislocation after diagnosis. The diabetes incidence was followed for 200 days.

### Insulitis scoring

The top, middle, and bottom of formalin-fixated paraffin-embedded pancreata were sectioned (5μm) from 10 GF and 10 STD mice at 13 weeks of age. Tissue was stained with haematoxylin and eosin. The degree of immune cell infiltration in the pancreatic islets (insulitis) was determined using a BX53 microscope at 20x (Olympus America, NY, USA) in a blinded manner using the following scoring system: 0, no infiltration; 1, intact islet with few mononuclear cells; 2, peri-insulitis; 3, islet infiltration <50%; 4, islet infiltration >50% (see [15]). 15–44 islets were scored per mouse (all visible islets were evaluated).

### Insulin autoantibodies (IAA)

Mice at ages 4-, 8-, and 13-weeks were fasted for 4 h and then anesthetized (4% isoflurane) during analgesia (IP morphine 0.04 mg in 100 μL sterile water). Blood was retrieved by heart puncture in EDTA-containing K2E tubes (BD, Franklin Lakes, NJ, USA). Tubes were inverted 10 times, immediately put on ice and centrifuged (10 min, 2000 g, 5°C) to retrieve plasma, which was stored at -80°C until analysis. IAA were measured using the Mouse Insulin Autoantibody ELISA kit (Abbexa Ltd, Cambridge, United Kingdom).

### Intraperitoneal glucose tolerance tests (IPGTT)

13–14-week-old mice were fasted for 6 h from 6 AM in individual cages with *ad libitum* access to water. Fasting blood glucose was measured followed by IP injection of 0.01 mL 1 mol/L glucose per g body weight and measurement of blood glucose at t = 5, 15, 30, 60, 90, and 120 min post glucose injection.

### Intraperitoneal insulin tolerance tests (IPITT)

13–14-week-old mice were fasted for 4 h from 8 AM in individual cages with *ad libitum* access to water. Fasting blood glucose was measured followed by IP injection of 0.00075 U insulin/g body weight (Actrapid®, Novo Nordisk, Bagsværd, Denmark) and measurement of blood glucose at t = 30, 60, 90, 120, 150, and 180 min post insulin injection.

### Statistics

Statistical analysis was performed in GraphPad Prism version 9.3.1 (La Jolla, CA, USA). The cumulative diabetes incidence in the two groups was compared using the Logrank Mantel-Cox test. Area under the curve (AUC) was calculated to compare IPGTT (with fasting blood glucose as baseline) and IPITT responses between groups. Comparisons between the groups were performed with a two-tailed un-paired t-test. If normal distribution or equal variance were not valid, a two-tailed Mann-Whitney $U$ test was used. $p$-values ≤0.05 were considered statistically significant. Data are shown as means ± SEM.

## Results

The GF diet provided strictly *in utero* compared to the STD diet showed no reduction in auto-immune diabetes incidence during 200 days of recording (50% GF vs 41% STD) (Fig 1a). Also, no difference in the average insulitis score at 13 weeks of age was observed between the mouse groups (1.88 GF vs 1.98 STD) (Fig 1b). Furthermore, insulitis scores (0, 2, 3, and 4) were equally distributed between the groups; however, a significantly higher distribution of insulitis score 1 (few mononuclear cells in an islet) was observed in GF mice compared to STD mice (236%, $p = 0.0032$, U = 14, 95.67% confidence interval (3.4%;7.7%)) (Fig 1c). The levels of plasma IAA did not differ between the mouse groups at 4-, 8-, and 13-weeks of age (Fig 1d).

The GF diet *in utero* had no effect on glucose (Fig 2a and 2b) nor insulin (Fig 2c and 2d) tolerance. Lastly, the GF diet contained 158.5 ppm (mg/kg) gliadin and the STD diet contained 27,675 ppm gliadin. The GF diet used in [11] contained 45.1 ppm (batch A) and 99.1 ppm gliadin (batch B).

## Discussion

Previously, we found that a GF diet provided strictly *in utero* compared to a lifelong gluten-containing STD diet alleviated autoimmune diabetes markedly in NOD mice. The present study could not replicate the diabetes alleviative effect behind the GF diet intervention as no differences was observed on incidence, insulitis score, titer of plasma IAA, and glucose and insulin intolerance in NOD mice fed a GF vs a STD diet *in utero*.

The GF diet used in the current study had a gliadin level on 158.5 ppm, based on measurement with the RIDASCREEN® Gliadin detection kit, which is above the U.S. Food and drug administration's limit on 20 ppm for GF products. Thus, the presence of gluten in the GF diet could potentially explain why we do not observe any diabetes alleviative effect behind the GF diet intervention. This prompted us to measure the gliadin content with the RIDASCREEN® Gliadin detection kit in 2 batches of GF diet used in our previous study that found a reduced insulitis score in NOD mice exposed to a GF diet *in utero* vs a lifelong STD diet [11]. Surprisingly, the analysis also showed presence of gliadin (45.1 and 99.1 ppm) in these batches, although previous analysis of the gliadin content in the GF diet formula with the GlutenTox ELISA Competitive assay (Biomedal Diagnostics, Sevilla, Spain) showed no presence of gluten

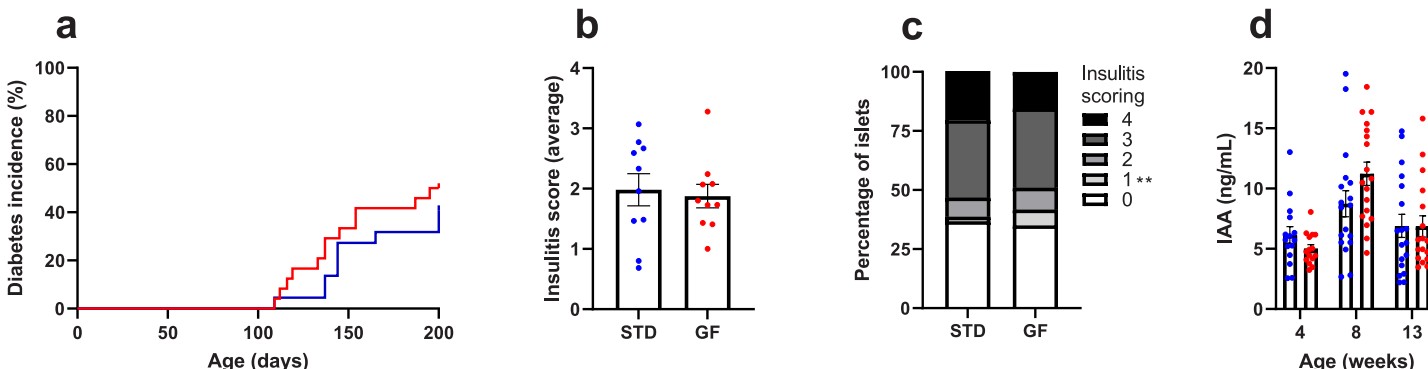

**Fig 1. No diabetes-alleviating effect of a GF diet strictly *in utero*. a)** cumulative diabetes incidence (STD n = 22, GF n = 24), **b)** mean relative insulitis score from 13-week-old NOD mice on a scale from 0–4 (STD n = 10, GF n = 10), **c)** mean insulitis scoring distribution in percentage from 13-week-old NOD mice on scale 0–4 (STD n = 10, GF n = 10), **d)** plasma IAA levels (STD 4, 8, 13 weeks n = 15, 18, 18; GF 4, 8, 13 weeks n = 15, 18, 17). Blue dots/line = STD NOD group, red dots/line = GF NOD group. Data are means ± SEM. Logrank Mantel-Cox test in **a)**. Un-paired two-tailed *t*-test or Mann-Whitney U-test in **b, c, d)**. **$p<0.01$. GF = NOD mice fed a gluten-free diet *in utero* and a STD diet from birth, IAA = insulin autoantibodies, NOD = Non-Obese Diabetic, STD = NOD mice fed a standard gluten-containing diet throughout life including *in utero*.

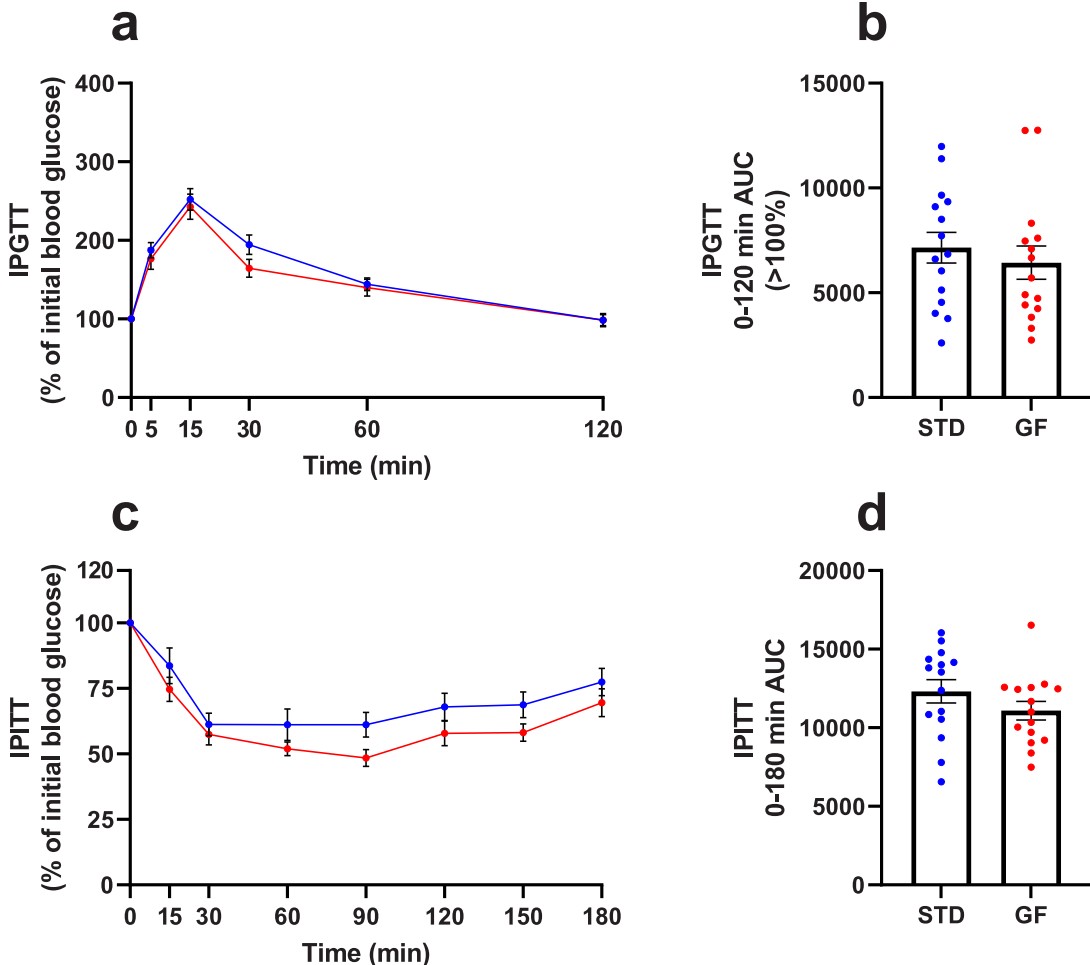

**Fig 2. Unimproved insulin and glucose tolerance from a GF diet strictly *in utero*. a)** IPGTT and **c)** IPITT on 13-14-week-old mice. **a)** 0.01 mL 1 mol/L glucose/g body weight (STD n = 15, GF n = 15) and **c)** 0.00075 U insulin/g body weight (STD n = 15, GF n = 14) was injected at t = 0 min and blood glucose levels were measured at indicated times. AUC was calculated to assess differences in **b)** glucose tolerance and **d)** insulin tolerance. Blue dots/line = STD NOD group, red dots/line = GF NOD group. Data are means ± SEM. Un-paired two-tailed *t*-test in **b, d**). AUC = area under the curve, GF = NOD mice fed a gluten-free diet *in utero* and a STD diet from birth, IPGTT = intraperitoneal glucose tolerance tests, IPITT = intraperitoneal insulin tolerance tests, NOD = Non-Obese Diabetic, STD = NOD mice fed a standard gluten-containing diet throughout life including *in utero*.

(<0.075 ppm) [10]. Thus, the exact gliadin content in diet can be difficult to measure and varies with the type of kit used, calling for a thorough comparative analysis of the available gliadin detection kits. Based on our comparative measurements with the RIDASCREEN® Gliadin detection kit, the missing diabetes-alleviative effect in the current study is likely not explained by an elevated gluten content in the GF diet.

The previous studies in our laboratory that investigated the effect of a GF diet were performed with NOD mice from Taconic Biosciences, whereas the mice in the current study were from the Jackson Laboratory. The choice of vendor has an impact on the microbiota composition, where different strains of mice cluster together in a vendor-dependent manner instead of a strain-specific manner [16], and modulating the microbiota influences diabetes development in NOD mice [17]. Furthermore, different breeding centers observe different diabetes incidences [18]. Thus, the inconsistencies between the current study and the

previous ones performed in our laboratory could be caused by vendor-specific microbiota differences in the NOD mouse colonies. On the other hand, a recent study showed a reduction in autoimmune diabetes incidence in antibiotic-treated NOD mice on a GF diet *in utero* and the weaning period compared to antibiotic-treated NOD mice on a STD diet throughout life [6]. This indicates that the diabetes-alleviating effects of GF diet is not entirely microbiota-derived.

Currently, it is not possible to decipher the exact cause of the discrepancies in the diabetes alleviative effect of the GF diet intervention between the current and previous studies. To our knowledge, Antvorskov et al. and Haupt-Jorgensen et al. are the only published studies that have examined the diabetes alleviating effect of a GF diet strictly *in utero* in NOD mice. Given the difficulties of replicating the results, it seems that the diabetes alleviating effect with this experimental set-up is not as consistent as previously believed and vary with other factors. This would match several human studies showing no association between islet autoimmunity and/or T1D development in offspring and maternal consumption of gluten-containing food products [19–22]. Still, a study from our laboratory based on questionnaire data from the Danish National Birth Cohort demonstrated an increased proportional risk of T1D in children per 10g/day increase of maternal gluten intake [23], suggesting that even lowering the intake of gluten maternally reduced the risk of T1D in the children.

In conclusion, autoimmune diabetes was not alleviated in NOD mice exposed to a GF diet *in utero* as previously shown. Based on animal intervention studies and epidemiological studies, we believe that this may be caused by mice being from different vendors and having different microbiota compositions in the compared studies. Hence, the mechanism behind the diabetes-alleviating effect of a GF diet *in utero*, which we intended to explore in this study, is dependent also on other factors. However, as a GF diet would be a relatively easy and safe preventive treatment for humans, the possible diabetes-alleviative effect of a GF diet should be further explored. Initially, this will require a large-scale animal study using NOD mice from different vendors, testing the possible diabetes-alleviative effects of a strictly GF diet initiated at different time points, including analyzing both the intestinal microbiota composition and histology at different prediabetic ages and the diet gliadin levels with a reliable analysis kit.

## Supporting information

**S1 Data.**
(XLSX)

## Acknowledgments

Authors thank Biomedical Laboratory Scientist Maria Louise Nielsen (The Bartholin Institute, Department of Pathology, Rigshospitalet) for excellent technical assistance.

## Author Contributions

**Conceptualization:** Mia Øgaard Mønsted, Karsten Buschard, Martin Haupt-Jorgensen.

**Methodology:** Mia Øgaard Mønsted, Laurits Juulskov Holm, Martin Haupt-Jorgensen.

**Writing – original draft:** Mia Øgaard Mønsted, Martin Haupt-Jorgensen.

**Writing – review & editing:** Mia Øgaard Mønsted, Laurits Juulskov Holm, Karsten Buschard, Martin Haupt-Jorgensen.

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
