## [Decision Letter · Decision Letter 0]

28 Feb 2023

PONE-D-23-01826

Missing diabetes alleviative effect of a maternal gluten-free diet in offspring of non-obese diabetic mice

PLOS ONE

Dear Dr. Haupt-Jørgensen,

Thank you for submitting your manuscript to PLOS ONE. After careful consideration, we have decided that your manuscript does not meet our criteria for publication and must therefore be rejected.

I am sorry that we cannot be more positive on this occasion, but hope that you appreciate the reasons for this decision.

Kind regards,

Takahiro Nemoto, Ph.D

Academic Editor

PLOS ONE

**Additional Editor Comments:**

This paper was fairly judged by two judges. Unfortunately, the results were judged to be insufficient for publication in this paper.

Reviewers' comments:

Reviewer's Responses to Questions

**Comments to the Author**

1. Is the manuscript technically sound, and do the data support the conclusions?

Reviewer #1: Partly

Reviewer #2: Yes

2. Has the statistical analysis been performed appropriately and rigorously? 

Reviewer #1: Yes

Reviewer #2: Yes

3. Have the authors made all data underlying the findings in their manuscript fully available?

Reviewer #1: Yes

Reviewer #2: Yes

4. Is the manuscript presented in an intelligible fashion and written in standard English?

Reviewer #1: Yes

Reviewer #2: Yes

5. Review Comments to the Author

Reviewer #1: For this current study, the authors tired to repeat their previous published results. However, the authors used a different animal vendor, a different GF food, and a different assay for gluten detection. They could not figure out the cause of the discrepancies in the diabetes alleviate effect of the GF diet intervention between the current and previous studies. Basically, this current study did not provide any useful information. Therefore, I would suggest to reject this manuscript.

Reviewer #2: This is a negative study showing that a gluten free (GF) diet during the pregnancy of NOD mice had no effect on the rate of diabetes development in 1st generation offspring. There were also no differences in glucose and insulin tolerance, insulitis scores, insulin autoantibody titres in the offspring of GF and standard mouse diet (STD) treated pregnant NOD mice.

It is important to report negative studies, so I applaud the authors for submitting this m/s for publication.

The NOD mice of this study were from the Jackson lab rather than from Taconic. The authors almost dismiss this as the possible reason for the negative result here. The mice of the two different sources are quite different genetically and this may well be the reason.

I have a few comments for improving the manuscript.

1) The title is difficult to follow. Suggest something like: Failure to replicate effects of a maternal gluten free diet in non-obese diabetic mice to reduce rates of offspring type 1 diabetes development

2) Line 59 - should be 'a multifactorial disease'.

3) Line 106 - "of 0.01 mL 0.00075 U insulin (Actrapid®, Novo Nordisk, Bagsværd, Denmark) per g body weight does not make sense" Should the '0.00075 U insulin be a concentration (units/violume)?

4) Through the results section there are formatting errrors in the PDF " (Error! Reference source not found.a). " These need to be corrected in a new version.

5) In results section Lines 121-123 "a significantly higher distribution of insulitis score 1 (few mononuclear cells in an islet) was observed in GF mice compared to STD mice (236%, p=0.0032, U=14, 95.67% confidence interval (3.4%;7.7%))" doesn't make sense to me. Is there a clearer way to present these results. It is also very unclear if there is any real biological significance of this result. Could insulitis Score 0 and 1 be combined?

6) In Figure 2, it would be better to present actual glucose levels rather than % from baseline as this is more informative and can allow comparison to other studies. Were mice that were already diagnosed with diabetes excluded from these tests??

6. PLOS authors have the option to publish the peer review history of their article (what does this mean?). If published, this will include your full peer review and any attached files.

Reviewer #1: No

Reviewer #2: No

- - - - -

---

## [Author Response · Author response to Decision Letter 0]

28 Mar 2023

Letter of response to the comments from the academic editor and reviewers

Ms. Ref. No.: PONE-D-23-01826

We wish to thank the editor and reviewers for the valuable and relevant comments and questions. We have made revisions of the manuscript (marked in red) in accordance with the reviewers’ suggestions. This letter outlines in detail the changes made in the manuscript, referring point-by-point to the questions/criticisms of the reviewers and the editors. 

Additional Editor Comments:

This paper was fairly judged by two judges. Unfortunately, the results were judged to be insufficient for publication in this paper.

AUTHOR RESPONCE:

Although we respect the editor’s opinion, we are puzzled about the reasons to reject and would like to challenge the decision with the following arguments:

1. Both reviewer 1 and 2 agree that the manuscript is technically sound and that the quality of the statistics, presentation of data, and language is high/sufficient.

2. Reviewer 2 applause us for submitting our manuscript reporting a negative finding. Moreover, reviewer 2 is overall positive and has suggestions for improving the manuscript that we have now implemented in the manuscript (se comments below). In our opinion, the changes have improved the manuscript. 

3. In our opinion reviewer 1 has unfortunately misunderstood a few central points. First, the purpose of our study was not to try and repeat the previous finding of a gluten-free diet having an alleviative effect for type 1 diabetes. The purpose was to further dissect the mechanisms behind. Moreover, to adequately show that a gluten-free diet alleviates diabetes it is in our opinion important to show that the effect is stable; hence, the effect should not be vendor specific etc. We strongly disagree with reviewer 1 that the study did not provide any useful information, as this statement seems purely to be based on the fact that we could not replicate previous findings. However, the study was conducted in exactly the same way and with the same high quality as our previous studies that found an effect of the gluten-free diet.

4. PLOS ONE encourage researcher to share their negative and null results: https://everyone.plos.org/2020/04/06/filling-in-the-scientific-record-the-importance-of-negative-and-null-results/

5. A gluten-free diet has show some anti-diabetogenic effects in both humans (interventions studies and epidemiological studies) as well as in mouse models. However, some studies do not find this effect, as we openly report in our manuscript. We know that other groups have had problems showing that a gluten-free diet reduces the autoimmune diabetes incidence in NOD mice but they have not published their work. Our collaborators studying the anti-diabetic effect of a gluten-free diet has encouraged us to publish our negative results. In our opinion, not publishing our result would basically be bad research practice and can potentially lead to the believe that a gluten-free diet is anti-diabetogenic, when it may not be or may be less anti-diabetogenic than we previously believed. Hence, not publishing these results could ultimately have consequences for the type 1 diabetes patients, which is why it is so important to get our results out for the type 1 diabetes community to decide. 

Comments from the Reviewer:

Reviewer #1: For this current study, the authors tired to repeat their previous published results. However, the authors used a different animal vendor, a different GF food, and a different assay for gluten detection. They could not figure out the cause of the discrepancies in the diabetes alleviate effect of the GF diet intervention between the current and previous studies. Basically, this current study did not provide any useful information. Therefore, I would suggest to reject this manuscript.

AUTHOR RESPONCE:

Thank you for the comments. The purpose of our study was not to try and repeat the previous finding of a gluten-free diet having an alleviative effect for type 1 diabetes. The purpose was rather to further dissect the mechanisms behind. Moreover, to adequately show that a gluten-free diet alleviates diabetes it is in our opinion important to show that the effect is stable; hence, the effect should not be vendor specific etc. We respectfully disagree with you that the study did not provide any useful information, as this statement seems to be based on the fact that we could not replicate previous findings. However, the study was conducted in exactly the same way, with the same diet recipe, and with the same high quality as our previous studies that found an effect of the gluten-free diet. Moreover, a gluten-free diet has shown some anti-diabetogenic effects in both humans (interventions studies and epidemiological studies) as well as in mouse models. However, some studies do not find this effect, as we openly report in our manuscript. We know that other groups have had problems showing that a gluten-free diet reduces the autoimmune diabetes incidence in NOD mice but they have not published their work. Our collaborators studying the anti-diabetic effect of a gluten-free diet has encouraged us to publish our negative results. In our opinion, not publishing our result would basically be bad research practice and can potentially lead to the believe that a gluten-free diet is anti-diabetogenic, when it may not be or may be less anti-diabetogenic than we previously believed. Hence, not publishing these results could ultimately have consequences for the type 1 diabetes patients, which is why it is so important to get our results out for the type 1 diabetes community to decide. 

Reviewer #2: This is a negative study showing that a gluten free (GF) diet during the pregnancy of NOD mice had no effect on the rate of diabetes development in 1st generation offspring. There were also no differences in glucose and insulin tolerance, insulitis scores, insulin autoantibody titres in the offspring of GF and standard mouse diet (STD) treated pregnant NOD mice.

It is important to report negative studies, so I applaud the authors for submitting this m/s for publication.

The NOD mice of this study were from the Jackson lab rather than from Taconic. The authors almost dismiss this as the possible reason for the negative result here. The mice of the two different sources are quite different genetically and this may well be the reason. Agree, we have added more about this possible reasons in the conclusion of the discussion.

I have a few comments for improving the manuscript.

1) The title is difficult to follow. Suggest something like: Failure to replicate effects of a maternal gluten free diet in non-obese diabetic mice to reduce rates of offspring type 1 diabetes development. AUTHOR RESPONCE: Agree, corrected.

2) Line 59 - should be 'a multifactorial disease'. AUTHOR RESPONCE: Agree, corrected.

3) Line 106 - "of 0.01 mL 0.00075 U insulin (Actrapid®, Novo Nordisk, Bagsværd, Denmark) per g body weight does not make sense" Should the '0.00075 U insulin be a concentration (units/violume)? AUTHOR RESPONCE: Agree, we have corrected to: 0.00075 U insulin/g body weight

4) Through the results section there are formatting errrors in the PDF " (Error! Reference source not found.a). " These need to be corrected in a new version. AUTHOR RESPONCE: Agree, corrected.

5) In results section Lines 121-123 "a significantly higher distribution of insulitis score 1 (few mononuclear cells in an islet) was observed in GF mice compared to STD mice (236%, p=0.0032, U=14, 95.67% confidence interval (3.4%;7.7%))" doesn't make sense to me. Is there a clearer way to present these results. It is also very unclear if there is any real biological significance of this result. Could insulitis Score 0 and 1 be combined? AUTHOR RESPONCE: We would like to stay with the current way of showing our results, as it is the usual way. We would rather not combine score 0 and 1, as these are different degrees of insulitis.

6) In Figure 2, it would be better to present actual glucose levels rather than % from baseline as this is more informative and can allow comparison to other studies. Were mice that were already diagnosed with diabetes excluded from these tests?? AUTHOR RESPONCE: We would prefer to stay with %, as our lab normally reports the data that way. This allows us/the readers to compare our current and previous results easily. Only normoglycemic mice were used for analysis, as now specified in the method section.

Yours sincerely, 

Martin Haupt-Jørgensen

---

## [Decision Letter · Decision Letter 1]

18 May 2023

PONE-D-23-01826R1Failure to replicate the diabetes alleviating effect of a maternal gluten-free diet in non-obese diabetic micePLOS ONE

Dear Dr. Haupt-Jørgensen,

Thank you for submitting your manuscript to PLOS ONE. After careful consideration, we feel that it has merit but does not fully meet PLOS ONE’s publication criteria as it currently stands. Therefore, we invite you to submit a revised version of the manuscript that addresses the points raised during the review process. Specifically, the authors should address the following points: (1) The suthors state that " The current study was initiated to further explore the mechanism behind the reported diabetes alleviating effect of a GF diet in utero". However, the used methodology are quite similar to that used in previous research. Therfore, the authors should explain how they would "further explore the mechanisms" using the same methods. Did the authors plan for more advanced or additional studies but they didn't conduct them given the "negative" results? If this is the case, authors should explain.(2) In the discussion, the authors provided some good possible explanations for why they coud not replicate the results of their previous studies. However, the whole manuscript gives the impression that the the positive findings of the authors' previous studies are the "correct", although it is possible that the "positive" findings in the previous studies could be due to type 1 error, and the "negative" findings in this study could be due to type 2 error.    (3) While the the authors provided some possible explanations for the inconsistent findings between this and the previous studies, they should elaborate on the implications of such findings on both the clinical and research levels with future researrch directions, particulary as the authors conclude that "the possible diabetes-alleviative effect of a GF diet should be further explored". For example, what precautions/points the researchers should be aware of when they perform similar studies in the future? should we do such study using mice from different vendors? Should we compare different gliaden detection kits? Should future studies assess microbiota composition? etc...  (4) Authors have to acknowldge some study limitations. For example, if the main aim of the study was to "further explore the mechanism behind the reported diabetes alleviating effect of a GF diet in utero", they should have strictly followed the same methods of the previous study, including mice from same vendors, although it could be an important finding that the vendor change could possibly affect the findings!. Second, it would be helpful to to do histopathoological studies in the intestine (like the previous studies). Third, assessment of microbiota might have provided useful information. (5) (Error! Reference source not found) still appears several times in the results section. 

We look forward to receiving your revised manuscript.

Kind regards,

Elsayed Abdelkreem, MD, PhD

Academic Editor

PLOS ONE

Journal Requirements:

2. Please amend either the title on the online submission form (via Edit Submission) or the title in the manuscript so that they are identical.

3. Please amend your authorship list in your manuscript file to include author Karsten Buschard.

4. Please ensure that you refer to Figures 1 and 2 in your text as, if accepted, production will need this reference to link the reader to the figure.

6. We notice that your manuscript file was uploaded on Jan 20 2023. Please can you upload the latest version of your revised manuscript as the main article file, ensuring that does not contain any tracked changes or highlighting. This will be used in the production process if your manuscript is accepted. Please follow this link for more information: http://blogs.PLOS.org/everyone/2011/05/10/how-to-submit-your-revised-manuscript/

Additional Editor Comments (if provided):

Reviewers' comments:

Reviewer's Responses to Questions

**Comments to the Author**

1. If the authors have adequately addressed your comments raised in a previous round of review and you feel that this manuscript is now acceptable for publication, you may indicate that here to bypass the “Comments to the Author” section, enter your conflict of interest statement in the “Confidential to Editor” section, and submit your "Accept" recommendation.

Reviewer #3: All comments have been addressed

Reviewer #4: (No Response)

Reviewer #5: All comments have been addressed

Reviewer #6: All comments have been addressed

2. Is the manuscript technically sound, and do the data support the conclusions?

Reviewer #3: Yes

Reviewer #4: Yes

Reviewer #5: Yes

Reviewer #6: Yes

3. Has the statistical analysis been performed appropriately and rigorously? 

Reviewer #3: Yes

Reviewer #4: Yes

Reviewer #5: Yes

Reviewer #6: Yes

4. Have the authors made all data underlying the findings in their manuscript fully available?

Reviewer #3: Yes

Reviewer #4: Yes

Reviewer #5: Yes

Reviewer #6: (No Response)

5. Is the manuscript presented in an intelligible fashion and written in standard English?

Reviewer #3: Yes

Reviewer #4: Yes

Reviewer #5: Yes

Reviewer #6: Yes

6. Review Comments to the Author

Reviewer #3: The authors present a very interesting manuscript that is very well written. I believe it is very important to publish this study as the authors show that the findings regarding alleviating diabetes with a gluten-free diet are not yet stable. Thus this study may have a large impact for patients that would be considering a gluten-free diet as an intervention. As these negative results may have great implications and it is in accordance with PLOS ONE recommendations to publish negative results, I recommend this manuscript for publication.

Reviewer #4: Monsted et al report the failure of reproducing earlier data obtained in NOD mice that have been fed a gluten-free diet in utero. The authors repeated experiments that have been published earlier in a very similar way with the goal to get a more mechanistic insight of how protection from T1D was achieved. Unfortunately, they have not been able to detect any differences between the experimental groups.

As an additional referee I share the opinion of the other two reviewers that the data are experimentally sound and the conclusions made are supported by the provided data. I also share the opinion of reviewer #2 that “negative” data should be published.

However, I also agree with the editor’s decision. Since the work demonstrated that data of an earlier publication of the same research group cannot be confirmed, the only sensible way to handle this dilemma would be to write a short retraction article to the initial published paper, send it to “Diabetes Metab Res Rev” and in fact retract the publication. Thereby the intention of the authors is irrelevant. Fact is that the present manuscript proves that the previous work was invalid and should therefore be retracted. The authors must admit that anything else would be unfair to the scientific community and the peer-review system since the authors would end up having two additional publications on their record that consist of very questionable data. However, once retracted, I suggest that the authors would publish the entire data set as “negative and null results”. This would also be in agreement with the author’s response #3 to reviewer #1, since the robustness of the study (i.e. food and mouse vendor) could be addressed properly.

Reviewer #5: The paper describes a failure to replicate the previously shown diabetes-alleviating effects of GF diet on NOD mice offspring while given in utero.

I agree with the authors and reviewer 2 that despite this being a negative study, the methodology is sound and the results are of importance. Failure to replicate results in dietary intervention studies increases current knowledge by highlighting the complexity through which diet can affect the outcomes. The authors discuss possible mechanisms which could have led to the current (negative) results and make effort to deliver explanations. The connection of GF diet and gut microbiota differences between vendors seems like a plausible explanation for the failure to replicate results. However, I think the future reader would appreciate if the authors were able to provide data or to explain (as a limitation) why it was not possible to examine gut microbiota composition of the mice from the current experiment (Jackson Laboratories) and compare them to the results of mice gut microbiota composition from previous successful experiments (Taconic). The possible observed differences might pinpoint the microbial taxa playing a role in relaying the diabetes alleviating effect of GF diet.

Reviewer #6: Even though this research produces negative data and contradicts previous results, I think this article deserves to be published with discussions in the article.

7. PLOS authors have the option to publish the peer review history of their article (what does this mean?). If published, this will include your full peer review and any attached files.

Reviewer #3: **Yes: **Martina Sebalo Vňuková

Reviewer #4: No

Reviewer #5: **Yes: **Vit Neuman, MD, PhD

Reviewer #6: No

---

## [Author Response · Author response to Decision Letter 1]

29 Jun 2023

Letter of response to the comments from the academic editor and reviewers

Ms. Ref. No.: PONE-D-23-01826R1

We wish to thank the editor and reviewers for the valuable and relevant comments and questions. We have made revisions of the manuscript (marked in red) in accordance with the reviewers’ suggestions. This letter outlines in detail the changes made in the manuscript, referring point-by-point to the questions/criticisms of the reviewers and the editors.

Specifically, the authors should address the following points:

(1) The suthors state that " The current study was initiated to further explore the mechanism behind the reported diabetes alleviating effect of a GF diet in utero". However, the used methodology are quite similar to that used in previous research. Therfore, the authors should explain how they would "further explore the mechanisms" using the same methods. Did the authors plan for more advanced or additional studies but they didn't conduct them given the "negative" results? If this is the case, authors should explain.

Anwser: Additional experiments were originally planned to explore the diabetes alleviative effect of the GF diet intervention, including analysis of microbial composition of cecal content, histological examination of different intestinal segments, IHC for inflammation markers in intestinal tissue and verification of microbial/inflammatory markers in plasma. These were, however, not performed due to the negative findings, which we write in the abstract, line 31-33: “In conclusion, this study could not replicate the previously observed diabetes alleviative effects of a maternal gluten-free diet in NOD mouse offspring and could therefore not further elucidate potential mechanisms.” and in the discussion, line 195-197: “Hence, the mechanism behind the diabetes-alleviating effect of a GF diet in utero, which we intended to explore in this study, is dependent also on other factors.”. Moreover, we have added the following sentence in the discussion, line 198-202: “Initially, this will require a large-scale animal study using NOD mice from different vendors, testing the possible diabetes-alleviative effects of a strictly GF diet initiated at different time points, including analyzing both the intestinal microbiota composition and histology at different prediabetic ages and the diet gliadin levels with a reliable analysis kit.”.

(2) In the discussion, the authors provided some good possible explanations for why they coud not replicate the results of their previous studies. However, the whole manuscript gives the impression that the the positive findings of the authors' previous studies are the "correct", although it is possible that the "positive" findings in the previous studies could be due to type 1 error, and the "negative" findings in this study could be due to type 2 error. 

Anwser: We have focused on laying forward our results in an unbiased way throughout the manuscript. However, we agree that the discussion could be more moderate and have thus now included/modified the following sentence in line 195-197: “Hence, the mechanism behind the diabetes-alleviating effect of a GF diet in utero, which we intended to explore in this study, is dependent also on other factors.”

(3) While the the authors provided some possible explanations for the inconsistent findings between this and the previous studies, they should elaborate on the implications of such findings on both the clinical and research levels with future researrch directions, particulary as the authors conclude that "the possible diabetes-alleviative effect of a GF diet should be further explored". For example, what precautions/points the researchers should be aware of when they perform similar studies in the future? should we do such study using mice from different vendors? Should we compare different gliaden detection kits? Should future studies assess microbiota composition? etc... 

Anwser: We agree on this. Hence, we have implemented the following sentence, line 198-202: Initially, this will require a large-scale animal study using NOD mice from different vendors, testing the possible diabetes-alleviative effects of a strictly GF diet initiated at different time points, including analyzing both the intestinal microbiota composition and histology at different prediabetic ages and the diet gliadin levels with a reliable analysis kit.

Moreover, for the question if we should compare different gliadin detection kits, also see line 163-165. Furthermore, see the following article comparing different gliadin detection analysis kits: ”Comparative study of commercially available gluten ELISA kits using an incurred reference material” Z. Bugyi et al 2013 Quality assurance and safety of crops and foods.

(4) Authors have to acknowldge some study limitations. For example, if the main aim of the study was to "further explore the mechanism behind the reported diabetes alleviating effect of a GF diet in utero", they should have strictly followed the same methods of the previous study, including mice from same vendors, although it could be an important finding that the vendor change could possibly affect the findings!. Second, it would be helpful to to do histopathoological studies in the intestine (like the previous studies). Third, assessment of microbiota might have provided useful information. 

Anwser: Thank you for the comment. We can confirm that the main aim was to further explore the mechanisms behind the GF diets diabetes alleviating effect. However, we strongly believed to get the same effect with NOD mice from a different vendor and it was moreover not possible at that point to get a sufficient number of mice from the original vendor that we bought the NOD mice from in our previous studies. These are the reasons why we got the mice from a different vendor. Also, to really trust the diabetes alleviating effect behind the GF diet, the effect should show stable results, including in NOD mice from different vendors, which was not the case, unfortunately. Besides using another vendor and another gliadin detection kit, the procedure used in our different GF intervention studies was perfectly the same. 

To address the raised concern from the reviewer, we have implemented the following sentence, line 198-202: Initially, this will require a large-scale animal study using NOD mice from different vendors, testing the possible diabetes-alleviative effects of a strictly GF diet initiated at different time points, including analyzing both the intestinal microbiota composition and histology at different prediabetic ages and the diet gliadin levels with a reliable analysis kit.

(5) (Error! Reference source not found) still appears several times in the results section. 

Anwser: Thank you. It could not be seen in the proof. We believe that it has been corrected now.

6. Review Comments to the Author

Reviewer #3: The authors present a very interesting manuscript that is very well written. I believe it is very important to publish this study as the authors show that the findings regarding alleviating diabetes with a gluten-free diet are not yet stable. Thus this study may have a large impact for patients that would be considering a gluten-free diet as an intervention. As these negative results may have great implications and it is in accordance with PLOS ONE recommendations to publish negative results, I recommend this manuscript for publication.

Anwser: Thank you for the comment.

Reviewer #4: Monsted et al report the failure of reproducing earlier data obtained in NOD mice that have been fed a gluten-free diet in utero. The authors repeated experiments that have been published earlier in a very similar way with the goal to get a more mechanistic insight of how protection from T1D was achieved. Unfortunately, they have not been able to detect any differences between the experimental groups.

As an additional referee I share the opinion of the other two reviewers that the data are experimentally sound and the conclusions made are supported by the provided data. I also share the opinion of reviewer #2 that “negative” data should be published.

However, I also agree with the editor’s decision. Since the work demonstrated that data of an earlier publication of the same research group cannot be confirmed, the only sensible way to handle this dilemma would be to write a short retraction article to the initial published paper, send it to “Diabetes Metab Res Rev” and in fact retract the publication. Thereby the intention of the authors is irrelevant. Fact is that the present manuscript proves that the previous work was invalid and should therefore be retracted. The authors must admit that anything else would be unfair to the scientific community and the peer-review system since the authors would end up having two additional publications on their record that consist of very questionable data. However, once retracted, I suggest that the authors would publish the entire data set as “negative and null results”. This would also be in agreement with the author’s response #3 to reviewer #1, since the robustness of the study (i.e. food and mouse vendor) could be addressed properly.

Anwser: Thank you for the comment. We do not believe that any methodological mistakes have been made in neither the current nor the previous published studies. Hence, we cannot see the idea in retracting the published articles. We believe that the scientifically correct way of handling the discrepant studies is to publish them, as the reality probably is that a gluten-free diet is not always diabetes alleviative i.e. less stable than previously believed. This is, however, important information that should not be held back. This opinion is moreover based on the human studies, both intervention studies and epidemiological studies that show inconsistent results. 

Reviewer #5: The paper describes a failure to replicate the previously shown diabetes-alleviating effects of GF diet on NOD mice offspring while given in utero.

I agree with the authors and reviewer 2 that despite this being a negative study, the methodology is sound and the results are of importance. Failure to replicate results in dietary intervention studies increases current knowledge by highlighting the complexity through which diet can affect the outcomes. The authors discuss possible mechanisms which could have led to the current (negative) results and make effort to deliver explanations. The connection of GF diet and gut microbiota differences between vendors seems like a plausible explanation for the failure to replicate results. However, I think the future reader would appreciate if the authors were able to provide data or to explain (as a limitation) why it was not possible to examine gut microbiota composition of the mice from the current experiment (Jackson Laboratories) and compare them to the results of mice gut microbiota composition from previous successful experiments (Taconic). The possible observed differences might pinpoint the microbial taxa playing a role in relaying the diabetes alleviating effect of GF diet.

Anwser: Thank you for the relevant comment. 

Other groups have found that a gluten-free diet alleviates type 1 diabetes in NOD mice: “A maternal gluten-free diet reduces inflammation and diabetes incidence in the offspring of NOD mice Hansen” CHF et. al. 2014. Hence, it seems that a gluten-free diet has some diabetes alleviate effect, but other factors such as vendor and microbiota affect the outcome. 

We agree that it would have been good to have done fecal microbiota analysis in our current and previous studies. However, in the previous studies the fecal microbiota composition was not analyzed, which was also the case in the current one. Hence, a comparison would not have been possible. Instead, in line 198-202, we now suggest that a new thorough study is initiated taking these limitations into considerations: “Initially, this will require a large-scale animal study using NOD mice from different vendors, testing the possible diabetes-alleviative effects of a strictly GF diet initiated at different time points, including analyzing both the intestinal microbiota composition and histology at different prediabetic ages and the diet gliadin levels with a reliable analysis kit.”

Reviewer #6: Even though this research produces negative data and contradicts previous results, I think this article deserves to be published with discussions in the article.

Anwser: Thank you for the comment.

Yours sincerely, 

Martin Haupt-Jørgensen

---

## [Decision Letter · Decision Letter 2]

17 Jul 2023

Failure to replicate the diabetes alleviating effect of a maternal gluten-free diet in non-obese diabetic mice

PONE-D-23-01826R2

Dear Dr. Haupt-Jørgensen,

We’re pleased to inform you that your manuscript has been judged scientifically suitable for publication and will be formally accepted for publication once it meets all outstanding technical requirements.

Kind regards,

Elsayed Abdelkreem, MD, PhD

Academic Editor

PLOS ONE

Additional Editor Comments (optional):

Reviewers' comments:

Reviewer's Responses to Questions

**Comments to the Author**

1. If the authors have adequately addressed your comments raised in a previous round of review and you feel that this manuscript is now acceptable for publication, you may indicate that here to bypass the “Comments to the Author” section, enter your conflict of interest statement in the “Confidential to Editor” section, and submit your "Accept" recommendation.

Reviewer #3: All comments have been addressed

Reviewer #4: (No Response)

Reviewer #5: All comments have been addressed

Reviewer #6: All comments have been addressed

2. Is the manuscript technically sound, and do the data support the conclusions?

Reviewer #3: Yes

Reviewer #4: (No Response)

Reviewer #5: Yes

Reviewer #6: Yes

3. Has the statistical analysis been performed appropriately and rigorously? 

Reviewer #3: Yes

Reviewer #4: (No Response)

Reviewer #5: Yes

Reviewer #6: Yes

4. Have the authors made all data underlying the findings in their manuscript fully available?

Reviewer #3: Yes

Reviewer #4: (No Response)

Reviewer #5: Yes

Reviewer #6: Yes

5. Is the manuscript presented in an intelligible fashion and written in standard English?

Reviewer #3: Yes

Reviewer #4: (No Response)

Reviewer #5: Yes

Reviewer #6: Yes

6. Review Comments to the Author

Reviewer #3: (No Response)

Reviewer #4: (No Response)

Reviewer #5: (No Response)

Reviewer #6: Thank you for improving the article according to suggestions and responding to reviewers. The article is better and can be published

7. PLOS authors have the option to publish the peer review history of their article (what does this mean?). If published, this will include your full peer review and any attached files.

Reviewer #3: No

Reviewer #4: No

Reviewer #5: No

Reviewer #6: No

---

## [Editor Report · Acceptance letter]

30 Aug 2023

PONE-D-23-01826R2 

Failure to replicate the diabetes alleviating effect of a maternal gluten-free diet in non-obese diabetic mice 

Dear Dr. Haupt-Jorgensen:

I'm pleased to inform you that your manuscript has been deemed suitable for publication in PLOS ONE. Congratulations! Your manuscript is now with our production department. 

Kind regards, 

on behalf of

Dr. Elsayed Abdelkreem 

Academic Editor

PLOS ONE